# The Complete Chloroplast Genome and the Phylogenetic Analysis of *Fimbristylis littoralis* (Cyperaceae) Collected in Cherry Blossom Nursery

**DOI:** 10.3390/ijms26052321

**Published:** 2025-03-05

**Authors:** Zhaoliang Gao, Yutong Cai, Jiaqi Long, Bo Wang, Zhaofeng Huang, Yuan Gao

**Affiliations:** 1Forest & Fruit Tree Research Institute, Eco-Environmental Protection Research Institute, Shanghai Academy of Agricultural Sciences, Shanghai 201403, China; gaozhaoliang@saas.sh.cn (Z.G.); 1476838839@nefu.edu.cn (Y.C.); 2School of Chemical and Environmental Engineering, Shanghai Institute of Technology, Shanghai 201418, China; 2110741114@mail.sit.edu.cn (J.L.); wangb@sit.edu.cn (B.W.); 3State Key Laboratory for Biology of Plant Diseases and Insect Pests, Institute of Plant Protection, Chinese Academy of Agricultural Sciences, Beijing 100193, China

**Keywords:** sedges, chloroplast genome, repeat, phylogenetic analysis, weed removal

## Abstract

*Fimbristylis littoralis*, also known as globe fringerush, is one of the most troublesome annual Cyperaceae weeds in dryland fields and nurseries in the Yangtze Plain, Middle and Lower in China. The chloroplast (cp) genome of *F. littoralis*, and even this genus, has not been studied yet. In this study, the feature of the cp genome of *F. littoralis* and its phylogenetic relationships has been reported for the first time. It exhibited a typical circular tetramerous structure, with 86 protein-encoding genes. There were 149 simple sequence repeats (SSRs) and 1932 long repeats (LRs) detected. The IR expansion and contraction revealed the uniqueness of *F. littoralis* because there is a special cross-boundary gene, *rps3*, located at the LSC/IRb junction. Phylogenetic and divergence time dating analysis showed the close relationship between *F. littoralis* and the genus *Cyperus*, as well as many evolutionary directions of *Cyperaceae* family plants. The most recommended chemical method for removing this weed from nurseries is to spray 13 g ai ha^−1^ (the amount of active ingredient applied per hectare) of saflufenacil before emergence or 7.5 g ai ha^−1^ of halosulfuron-methyl after emergence. In conclusion, this study was the first to report the complete cp genome of a plant in the genus *Fimbristylis*. Our findings also provided valuable biological information for studying the phylogenetic relationships and evolution among the family *Cyperaceae*.

## 1. Introduction

Cyperaceae, commonly known as sedges, is one of the most common plants on Earth and is distributed in a wide range of regions, from tropical to temperate [1,2]. Cyperaceae has more than 80 genera and 4000 species worldwide, and there are about 500 species of 28 genera in China, including perennial and annual plants [3]. It is precisely because of the richness of plant species in Cyperaceae, global distribution characteristics, lineage diversity, broad ecological preferences, and adaptation to various origins, including C4 photosynthesis and whole center chromosomes, that the Cyperaceae is an ideal model family for studying plant evolution [4]. Furthermore, many species in this family have invaded farmland, posing a threat to crop growth. These plants entering rice fields seriously affects the yield and quality of rice; if there were 100–200 sedges per square meter of paddy field, this would lead to a 53–65% reduction in rice yield [5].

*Fimbristylis littoralis*, also known as globe fringerush, belonging to the genus *Fimbristylis*, is one of the most common annual Cyperaceae weeds in paddy fields or dryland nurseries. According to reports, globe fringerush is native to tropical America and is a very serious weed in South Asia, East Asia, and the Caribbean [6,7]. Each globe fringerush plant is capable of producing more than 10,000 seeds, and its root system is significantly more developed than cultivated crops, rendering it a highly detrimental weed in agricultural fields [8]. According to the research, the adaptive light and warm temperature will promote the germination of the seeds of globe fringerush; the seeds sown on the soil surface have the highest emergence rate, while the seeds buried in the soil at 1 cm depth do not germinate; the sensitivity of globe fringerush to salt stress and water stress was not high during germination [7]. Water saturation conditions favor globe fringerush emergence and rapid growth; unless the water layer of 5–10 cm is maintained for more than 14 days, floods will not affect the emergence of globe fringerush [9]. Certain populations of globe fringerush have developed resistance to herbicides with diverse modes of action, including Acetolactate Synthase (ALS) inhibitors and auxin-type herbicides, presenting new challenges for effective weed management strategies [10,11,12]. From the perspective of plant resource utilization, the methanol extract of globe fringerush has antipyretic and anti-nociceptive effects on mice [13].

Chloroplasts (cp) are essential organelles in photosynthetic plants and algae, responsible for conducting photosynthesis [14]. Chloroplasts contain genetic material, and their genomes are highly conserved owing to the lack of recombination, haploidy, and uniparental inheritance. As such, they can provide very rich evolutionary information [15,16,17]. In addition, the cp genome is small and easy to obtain completely compared to the nuclear genome; therefore, it has unique research value in phylogeny, species identification, and population genetics [18]. Due to these distinctive characteristics, the determination and analysis of ribosome organization within the chloroplasts system have emerged as crucial mechanisms for elucidating plant phylogeny and evaluating biodiversity [19,20,21]. In general, the cp genome has a typical quadripartite structure [22,23,24], and its circular structure is organized into large single copy (LSC) and small single copy (SSC) regions, which are separated by a pair of inverted repeat (IR) sequences, designated as IRa and IRb, with opposite orientations [16,25,26]. Sequences between IRa and IRb regions can generate triggered flip-flop recombination, thereby stabilizing single-copy regions [27]. Previous studies have demonstrated that the cp genome has been found to be particularly valuable for investigating the phylogeny and evolutionary history of most plant lineages, especially in the context of reticulate evolution (e.g., hybridization) and polyploidy [28,29,30]. With the advancement of cp genome-sequencing technology and the in-depth understanding of the cp genome by researchers, the genetic relationship of multiple species, such as the genus *Echinochloa*, *Ammannia*, *Camellia*, *Taxodium*, *Pterocarpus*, etc., has been uncovered [18,27,31,32,33]. To date, the information about the composition and structure of globe fringerush and the evolutionary relationships of common sedge species based on cp genome is still limited.

The primary objective of our study was to conduct a comprehensive analysis and report the cp genome of globe fringerush collected from a cherry nursery, marking the first such effort for this species. Furthermore, we aimed to perform a systematic evolutionary analysis of all cp genomes of major sedge species registered in the NCBI database, a pioneering endeavor in this field. By achieving these goals, our research will provide a theoretical foundation for understanding the regeneration of diversity and the utilization of resources within sedge species, thereby contributing valuable insights to the broader scientific community.

## 2. Results

### 2.1. Chloroplast Genome Component

The cp genome libraries of globe fringerush were constructed using the Illumina TruSeq™ Nano DNA Sample Prep Kit (Illumina, San Diego, CA, USA). In total, after trimming low-quality fragments in the raw data, 80,992,456 clean reads were mapped to the complete genome of globe fringerush. This sequencing has also reached sufficient sequencing depth (Appendix A). The raw reads were deposited in the NCBI GenBank database (PRJNA1206234). De novo assembly using NOVOPlasty v4.2 software (https://github.com/ndierckx/NOVOPlasty, accessed on 2 July 2024) resulted in circular genomes of 201,922 bp length with 33.45% GC content. The complete cp genomes displayed the typical quadripartite structure of most angiosperms, including LSC, SSC, and a pair of IRs (IRa and IRb) (Figure 1). The cp genome contained 135 genes, including 86 protein-coding genes (with a total length of 63,036 bp), 41 transfer RNA (tRNA) genes, and 8 ribosomal RNAs (rRNA). There were 63 protein-coding and 25 tRNA genes located within the LSC; 19 protein-coding, 11 tRNA-coding, and 4 rRNA-coding genes located within IRb or IRa; and 6 protein-coding and 2 tRNA genes located within the SSC. The total length of all non-coding RNA genes was 8980 bp. Repeat sequences were also detected in the cp genome of globe fringerush. A total of 149 simple sequence repeats (SSRs) were identified, including 23 distributed in the coding region. According to the number of repeated bases, the type of SSR was shown in Figure 2a. There were 1932 long repeats (LR) in the cp genome of globe fringerush. According to the Hamming distance (HD), the type of LR was shown in Figure 2b.

### 2.2. Gene Function Annotation and Classification

All genes in the cp genome of globe fringerush were functionally annotated and mainly belonged to the photosynthesis and self-replication categories, with 44 genes and 57 genes, respectively. The gene names, groups, and categories are listed in Table 1. In the globe fringerush cp genome, 85 genes were matched to the Non-Redundant Protein Sequence Database (NR), 60 to Gene Ontology (GO), 57 to Clusters of Orthologous Groups (COG), 76 to the Kyoto Encyclopedia of Genes and Genomes (KEGG), and 84 to Swiss. Among these genes, 42 could be matched to all five databases; these cross-relationships are shown in Figure 3a. Genes matched to GO were further classified as biological process (BP), cellular component (CC), and molecular function (MF), with most genes classified as BP (Figure 3b). Genes matched to KEGG were mainly involved in energy production and conversion, translocation, ribosomal structure and biogenesis, and transcription pathways (Figure 3c).

### 2.3. IR Expansion and Contraction

To further observe the potential expansion and contraction of the IR regions, gene variations at the IR/SSC and IR/LSC boundary regions of 17 sedge species were compared (Figure 4). The gene *rps3* crosses the junction of LSC and IRb with 377 bp in LSC and 355 bp in IRb. The *rps3* gene, as a boundary gene, only appears in globe fringerush, *Cyperus aromaticus*, *C. dysplasia*, *C. flavidus*, and *C. rotandus*, and it only crosses the junction of LSC/IRb in the cp genome of *F. littoralis*. The gene *rpl22* is located in IRb, 405 bp away from the junction of LSC/IRb. This gene, as a boundary gene of the IR region, only appears in globe fringerush, *Carex giraldiana*, *C. microglochin*, and *C. capillifolia*. The gene *ndhG* located in the IRb region and the gene *ndhE* located in the SSC region are the boundary genes of SSC/IRb, and neither of them crosses the junction. The *ndhG* gene within the IR region, as a boundary gene, is very common in this family of plants, except for *C. amuricus* and *C. iria* in this study. The *ndhF*/*ndhG* genes were located in the junctions of SSC/IRa, which frequently appear as boundary genes at this location in plants of this family. The *rpl22*/*trnH* genes were located in the junctions of IRa/LSC. It is also common for the gene *trnH* to appear as a boundary gene at this position in plants of this family.

### 2.4. Phylogenetic Analysis and Divergence Time Dating Among Common Sedges

Phylogenetic trees were generated using maximum likelihood (ML) and Bayesian inference (BI) analysis methods based on 18 complete cp genomes showing the same topology (Figure 5). Cyperaceae plants clustered into a clade, which exhibited a further genetic relationship among this family. Globe fringerush and the genus *Cyperus* formed a monophyletic group with high support (support value (BS) = 100 for ML) (Figure 5a). It is estimated that they began to diverge 24.5 million years ago (Figure 5b). The genus *Carex* has a slightly distant phylogenetic relationship (BS = 100 for ML), which began to differentiate approximately 36.9 million years ago (Figure 5). These above-mentioned species share a common ancestor with *Gahnia tristis* and *Hypolytrum nemorum* (BS = 100 for ML), which began to differentiate 53.2 million years ago and 57.2 million years ago, respectively (Figure 5). The most distantly related species is *Scleria parvula*, which evolved into an independent branch as early as 72.6 million years ago (Figure 5).

### 2.5. Chemical Control Method

In pre-emergence application, as the dosage of halosulfuron-methyl increased, the germination inhibition rate of globe fringerush was significantly increased on the 21st day after herbicide treatment. However, even under the treatment of the recommended dosage (30.0 g ai ha^−1^ (the amount of active ingredient applied per hectare)) of halosulfuron-methyl, the germination inhibition rate of weeds is only 50.62%. On the contrary, at a quarter of the recommended dosage, 13 g ai ha^−1^ of saflufenacil can almost completely inhibit the emergence of globe fringerush (Figure 6a).

With post-emergence application, with the increase in the dosage of halosulfuron-methyl, saflufenacil, glyphosate, and fluroxypyr, globe fringerush was significantly inhibited on the 21st day after herbicide treatment and showed a decreasing gradient thereafter. Safufenacil and glyphosate cannot suppress the fresh weight of globe fringerush by more than 90% at the recommended dosage. A quarter of the recommended dosage of halosulfuron-methyl (7.5 g ai ha^−1^) and a half of the recommended dosage of fluroxypyr (90.0 g ai ha^−1^) can almost completely inhibit the growth of globe fringerush (>98%), respectively (Figure 6b).

## 3. Discussion

Cyperaceae plants are generally considered the third group of weeds, besides Gramineae and dicotyledonous weeds. In recent years, the rampant proliferation of such weeds in orchards, nurseries, and non-cultivated lands has become a highly distressing issue. The study of Cyperaceae plants has attracted increasing attention. The globe fringerush is a representative plant of the genus *Fimbristylis* in this family. The harm and herbicide resistance of globe fringerush have been reported frequently [7,8,9,11,12]. Not just a weed, globe fringerush also has a potential practical value. Research has shown that globe fringerush has allelopathic effects, not only on plants but also on microorganisms [34,35], which indicates that the extract of the plant has the potential to be developed as a natural product in integrated pest control strategies. There are also studies indicating that globe fringerush contain potential bioactive compounds with antioxidant, anticancer, and hepatoprotective effects [6,36], which indicates that the plant also has certain medicinal potential. However, the cp genome characteristics and phylogenetic studies based on them are still a research gap in this species and even in this genus. Since the cp genome sequence of tobacco was first reported [37], dozens of cp genomes from the genus *Carex*, *Gahnia*, *Hypolytrum*, *Scleria*, and *Cyperus* have been published, currently [38], which provides convenience for the study of the phylogenetic relationships and evolution of this family. As a common and harmful weed and also a plant with potential for utilization, the cp genome and phylogenetic information of globe fringerush are worth exploring deeply.

The globe fringerush was found to have an even larger cp genome than those of plants in the family Cyperaceae [38] because its size exceeds 20,000 bp (Figure 1). The typical circular tetramerous structure of the cp genome is conserved in plants [22,23,24], and the length of each quadripartite structure of the cp genome in the same genus is generally similar [31,39]. The results of our study confirmed the above notion (Figure 1). Simple sequence repeats, or microsatellites, are tandem repeats consisting of 1–6 nucleotide repeat units that are widely distributed in plant cp genomes [40,41]. As valuable molecular genetic markers, SSRs are widely used in plant genotyping and population genetics [42,43,44,45]. These repeats promote intermolecular recombination and enrich the diversity of cp genomes in the population [46]. All SSRs in the cp genome of globe fringerush were detected in this study (Figure 2a, Appendix A). Thus, differential SSRs can be used as important molecular markers in this species. Additionally, long repeats are special DNA sequences that are repeated in the genome in various forms and usually occupy a large proportion of the genome [47]. Repeated segments also have important molecular significance in the study of plant evolution [48]. Unlike SSRs, the cp genome of globe fringerush contains a large number of LRs, reaching 1932 (Figure 2b, Appendix A). The repeat sequences detected in this study are important biological information resources and are of great significance for the identification of globe fringerush and the study of genetic diversity and population structure.

Chloroplast genome genes are highly conserved in plants [15,16,17]. As previously mentioned, the cp genome of globe fringerush is relatively large (Figure 1), but there are only 63 protein-coding genes identified, showing a lower abundance than some weed plant cp genomes [32,33,49]. Although the genes were not completely consistent, the categories of genes were the same, mainly belonging to the categories of photosynthesis and self-replication, which further verifies the conservation of protein-encoding genes in chloroplasts [18,39,50]. The functional or pathway classification of these cp genes is also conservative [32,33,49]. It was reported that the plastid *clpP1* protease gene is essential for plant development [51]. Notably, the *clpP1* detected in the globe fringerush chloroplast may provide valuable evidence regarding the molecular evolution of this species. The relationship and mechanism between this gene and the development of globe fringerush deserve further research.

Expansion and contraction of the cp genome is a common phenomenon in plants [16], which occurs mainly at the IR/SC junction [52]. Despite their conserved nature, the dynamic processes of IR expansion and contraction significantly influence cp genome reorganization and size variation, representing a crucial mechanism in plant genome evolution [53,54,55,56]. This phenomenon has been observed in many plants [18,27,31]. This study showed that the IR expansion and contraction of the cp genome in the Cyperaceae plants, including globe fringerush. A new important discovery is that there is a unique cross-boundary gene, *rps3*, at the junction of LSC and IRb in the globe fringerush. Due to the fact that the cp genome of the genus *Fimbristylis* has never been reported, the IR expansion and contraction of globe fringerush in this study are not relatively similar to those in plants of this family. This is mainly manifested at the junction of LSC and IR, especially with a unique cross-boundary gene, *rps3*. The boundary genes at SSC/IRb and SSC/IRa are the same in many *Cyperus* species, reflecting the conservation of this family of plants. We also noticed that there is significant variation between sedge plants at the junction of LSC/IRa and LSC/IRb, which is reflected in differences in the types of boundary genes and differences in the length and distance of the same genes (Figure 4).

Genomic information is valuable for addressing species definitions, as this can establish organelle-based “barcodes” for certain species, which can then be used to reveal phylogenetic relationships [57]. Chloroplast genomic data serve as crucial molecular markers for species identification, phylogenetic reconstruction, and taxonomic classification within plant systems [58,59,60]. With the continuous discovery of plant cp genome information, the genetic evolutionary relationships of many plants have been successfully elucidated in the form of phylogenetic trees [18,39,50]. However, the phylogenetic relationships of the family Cyperaceae have not yet been reported. In the present study, the cp genomes of the common plants of the family Cyperaceae were performed phylogenetic analysis and divergence time dating estimation. Our research results clearly demonstrate the close relationship between globe fringerush and the plants in this family; that is, it has a closer relationship with the genus *Cyperus*. However, their divergence time was as early as 24.5 million years ago. Additionally, among the selected plants in this family, *Hypolytrum nemorum* and *Scleria parvula* have the earliest divergence time and the farthest phylogenetic relationship with other plants. While comprehensive analysis of complete chloroplast genomes may not fully resolve all phylogenetic relationships [59,61,62], it remains a valuable approach for elucidating interspecific relationships within this plant family.

Generally, economical and broad-spectrum herbicides will be used in seedling cultivation sites through directional spray. However, as pesticide pollution receives increasing attention [63,64], the strategy of reducing herbicide use and the precise use of herbicides are highly advocated [65,66]. However, globe fringerush is a relatively difficult weed to control in the Cyperaceae family, often requiring the use of large amounts of herbicides. To address this gap, we tested the removal efficacy of different herbicides before and after emergence. Based on our results, we strongly recommend using saflufenacil for pre-emergence management in nurseries where globe fringerush is expected to occur because this approach can achieve complete removal with only a quarter of the recommended dosage. If post-emergence treatment is necessary, the recommended herbicide to use is halosulfuron-methyl. Additionally, it is worth mentioning that this weed is prone to developing resistance, leading to ineffective herbicide control [10,11,12]. Therefore, we recommend conducting a pre-emergence treatment to achieve the best control effect with a low herbicide dosage. The sustainable management approach is certainly based on ecological strategies or even the resource utilization of the plant, but this still requires long-term research in the future.

## 4. Materials and Methods

### 4.1. DNA Extraction and Sequencing

Fresh leaves and stems were harvested from globe fringerush from a cherry blossom nursery in Shanghai (N 30.95°, E 121.48°), China, in 2022. Total genomic DNA was isolated using an optimized cetyltrimethylammonium bromide (CTAB) protocol, followed by the preparation of a 500 bp paired-end library with the NEBNext Ultra DNA Library Prep Kit (NEB, Ipswich, MA, USA) for subsequent Illumina sequencing. Sequencing was conducted on an Illumina NovaSeq 6000 platform (BIOZERON Co., Ltd., Shanghai, China). Approximately 8 GB of raw data from the globe fringerush, respectively, were generated with 150 bp paired-end read lengths.

### 4.2. Genome Assembly

Since the genomic information of this species has not yet been reported, a de novo assembly strategy was adopted for this sequencing [67]. The cp genome was assembled de novo using NOVOPlasty, with reference to the cp genomes of closely related species, resulting in two candidate circularized contigs. The contig exhibiting the highest cpDNA homology was selected as the candidate cp genome. Potential cp reads were identified from the Illumina read pool through BLAST version 2.15.0 alignment against both the NOVOPlasty-derived cp genome and the cp genomes of related species. These chloroplast-specific reads were subsequently utilized for de novo assembly of the cp genome using the SPAdes-3.13.0 software package. The NOVOPlasty assembly contig was optimized by the scaffolds from the SPAdes-3.13.0 result and aligned with the original clean Illumina reads using BWA, and base correction was performed with Pilon v1.22. Finally, the assembled sequence was reordered and oriented according to the reference cp genome to generate the final assembled cp genomic sequence.

### 4.3. Genome Component Analysis

Genes encoding proteins, tRNAs, and rRNAs in the cp genomes of globe fringerush were predicted using the GeSeq software (https://chlorobox.mpimp-golm.mpg.de/geseq.html, accessed on 2 July 2024). The specific parameters were set as follows: protein search identity: 60 [68]; rRNA, tRNA, DNA search identity: 35 [69]; 3rd party tRNA annotators: tRNAscan-SE v2.0.7. High-accuracy gene bundles were obtained by removing the redundancy of predicted initial genes, followed by manual correction of the head, tail, and exon/intron boundaries of the genes. Finally, for the base composition of the cp genome, the gene distribution of each interval, including LSC, SSC, and IR, and the classification of each functional gene were counted and summarized.

### 4.4. Gene Function Annotation and Classification Analysis

The protein sequences of cp genes were compared with known protein databases using BLASTP (evalue < 1 × 10^−5^) [70]. Because there may have been more than one alignment result for each sequence, only one optimal alignment result was reserved as the database alignment information of the gene to ensure its biological significance. These databases included the Non-Redundant Protein Sequence Database (NR) (http://www.ncbi.nlm.nih.gov/, accessed on 2 July 2024), Swiss-Prot (http://www.ebi.ac.uk/uniprot, accessed on 2 July 2024), Clusters of Orthologous Groups (COG), Kyoto Encyclopedia of Genes and Genomes (KEGG) (http://www.genome.jp/kegg/, accessed on 2 July 2024), and Gene Ontology (GO) (http://geneontology.org/, accessed on 2 July 2024). The amino acid sequences of globe fringerush were aligned with the above databases to obtain functional annotation information for the coding genes.

### 4.5. Contraction and Expansion Analysis of Inverted Repeats Regions

In addition to the newly sequenced cp genome of globe fringerush, sixteen cp genomes of common Cyperaceae plants (Appendix A) were downloaded from NCBI to perform the IR analysis. The quadripartite structure of each cp genome, comprising the large single-copy (LSC) region, small single-copy (SSC) region, and two inverted repeat (IR) regions, was systematically compared. Additionally, the effects of IR boundary shifts, including gene copy number variations and pseudogene formation resulting from IR expansion or contraction, were thoroughly analyzed [71]. Genes that crossed or were closest to the boundary were obtained. The function, length, and distance from the boundaries of these genes were analyzed.

### 4.6. Phylogenetic Analysis

In addition to the newly sequenced cp genome of globe fringerush, seventeen cp genomes of common Cyperaceae plants (Appendix A) were downloaded from the NCBI database and were used to resolve the cp phylogenetic tree. The sequences were aligned using ClustalW (v2.0.12) with the default settings. The DNA substitution model was assessed using the Akaike information criterion method [72]. The phylogenetic analysis was conducted using the maximum likelihood (ML) method implemented in PhyML v3.0 (htp://ww.atgc-montpeller.fr/phyml/, accessed on 17 December 2024), and the bootstrap was 1000 [73,74]. Bayesian inference was performed using MrBayes 3.1.2, following the methodology described by Wu et al. [75].

### 4.7. Divergence Time Dating Analysis

The time of divergence within the Cyperaceae (Appendix A) was estimated using BEAST v2.6.2 software [76,77]. The general time reversible (GTR) model was selected for site-specific evolutionary rate analysis. The log files were imported into Tracer v1.7.2 to compare the applicable molecular clock models. The coefficient of variation approaching 0 under the uncorrelated relaxed lognormal clock model suggested that the strict molecular clock model represented a more appropriate choice for the analysis. The Yule speciation model was utilized as the prior for tree estimation, with Markov chain Monte Carlo (MCMC) analyses run for 10 million generations. The convergence and stability of the MCMC results were evaluated using Tracer v1.7.2 [78], with all effective sample size (ESS) values exceeding 200, indicating satisfactory parameter estimation and model convergence. The maximum clade credibility (MCC) tree, incorporating species divergence time estimates, was generated using TreeAnnotator and subsequently visualized through tvBOT software version 1.0.0 [79].

### 4.8. Detection of Sensitivity to Common Herbicides

Globe fringerush seeds were sown in plastic pots (5 × 5 × 5 cm) containing middle-loam-type soil from farmland in the suburbs of Shanghai, China, where herbicides had never been used. Globe fringerush was cultivated to the 3.0-leaf stage outdoors after thinning (10 plants were retained) to prepare for post-emergence herbicide treatment. For the herbicide treatment group of pre-emergence treatment, soil spray was conducted directly after sowing. The herbicide spraying was conducted using an HCL2000 walking-type spraying system (Hengchuangli Intelligent Equipment Technology Co., Ltd., Suzhou, China); each treatment was performed with 50 mL of liquid (450 L ha^−1^ water) using a fan-shaped nozzle. The spray height was set to 35 cm and the walking speed of the nozzle was set to 240 mm/s. After 12 h of spraying, the pots were placed outdoors for cultivation. After 21 d of pre-emergence treatments, the number of grass seedlings was investigated. After 21 d of post-emergence treatments, the fresh weight of weeds was investigated. Subsequently, the inhibition rate of the number of grass seedlings and the aboveground fresh weight was calculated. The experiment contained at least three biological replicates, and the entire experiment was repeated twice.

All studies were conducted using the inhibition rate (IR) of the number of seedlings or fresh weight based on the values of CK. The experimental groups were randomly arranged. The data were subjected to ANOVA. To compare the differences in the percentage of inhibition rate, Duncan’s multiple range test (*p* < 0.05) was used. ANOVA was performed using SPSS version 20 (SPSS, Chicago, IL, USA).IR = (I_CK_ − I_T_)/I_CK_ × 100%
where IR represents inhibition rate of emergence or fresh weight, I_CK_ represents the emergence or fresh weight of the plants in the untreated group, and I_T_ represents the emergence or fresh weight of the plants in the treatment group.

## 5. Conclusions

This study is the first to report the feature of the cp genome of *F. littoralis* (globe fringerush) (even the first plant of the genus *Fimbristylis*) and comprehensively display the phylogenetic relationships among Cyperaceae. It was to provide a detailed introduction to the size, structure, and composition of the cp genome of globe fringerush. The phylogenetic analysis clearly shows the close relationship between globe fringerush and the genus *Cyperus*. Finally, we explored the best chemical strategy for the globe fringerush removal in the targeted spray area.

## Figures and Tables

**Figure 1 ijms-26-02321-f001:**
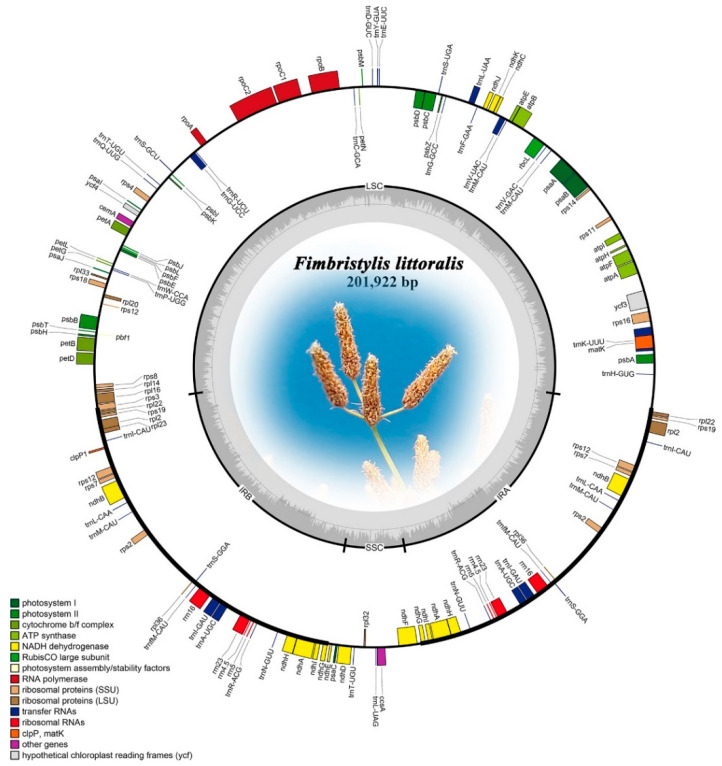
The assembly, size, and features of the chloroplast genome of *Fimbristylis littoralis*.

**Figure 2 ijms-26-02321-f002:**
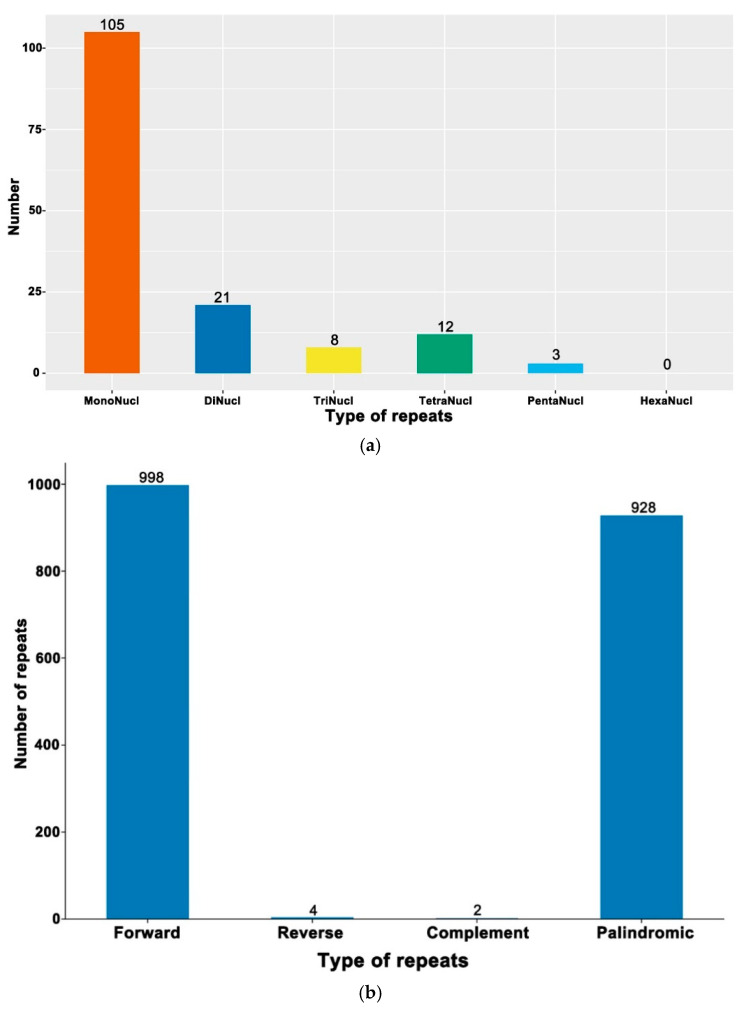
The repeat sequences in the chloroplast genome of *Fimbristylis littoralis*. (**a**) Simple sequence repeats. (**b**) Long repeats.

**Figure 3 ijms-26-02321-f003:**
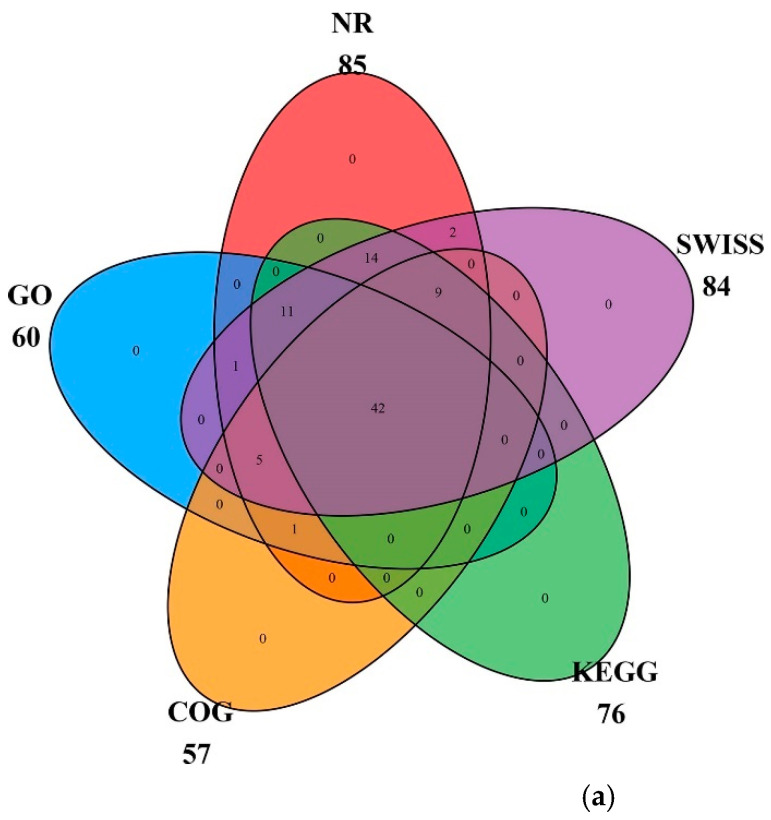
Classifications of genes function of *Fimbristylis littoralis*. (**a**) Cross matching of coding genes to five databases. (**b**) Percentages of genes matched to GO function classification. BP means biological process, CC means cellular component, and MF means molecular function. (**c**) Top ten of unigenes matched to KEGG pathways.

**Figure 4 ijms-26-02321-f004:**
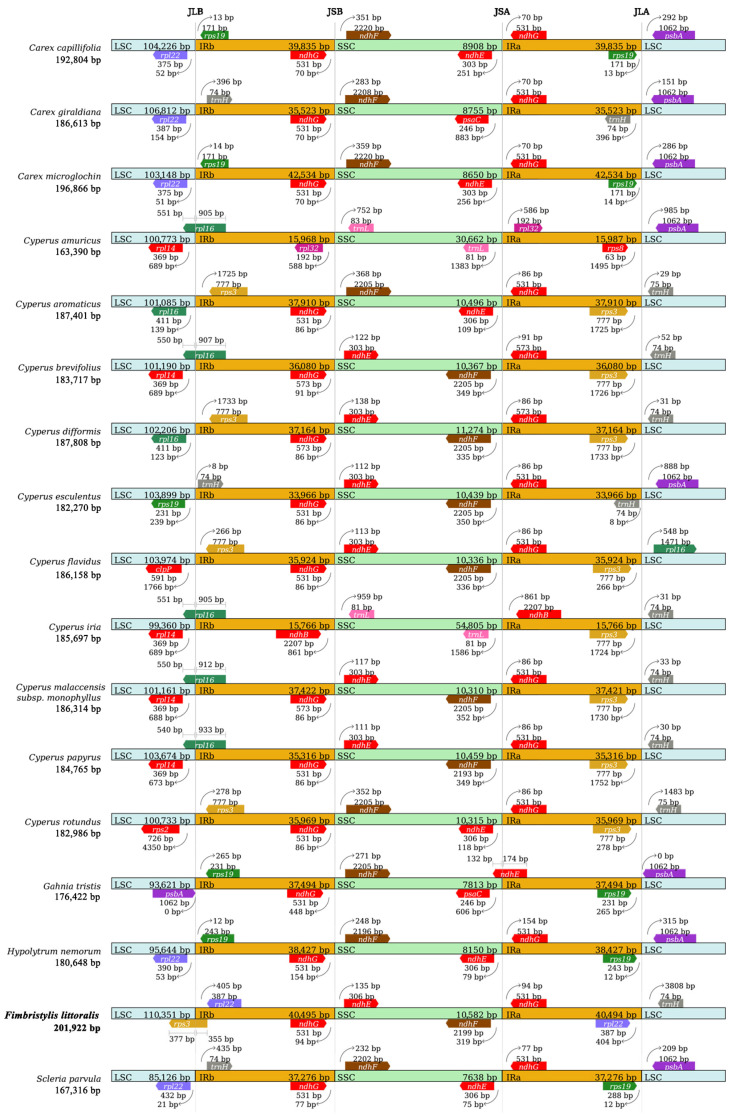
Comparison of large sequence copy (LSC), inverted repeat (IRb, IRa), and small sequence copy (SSC) border regions of the chloroplast genomes of representative Cyperaceae plants.

**Figure 5 ijms-26-02321-f005:**
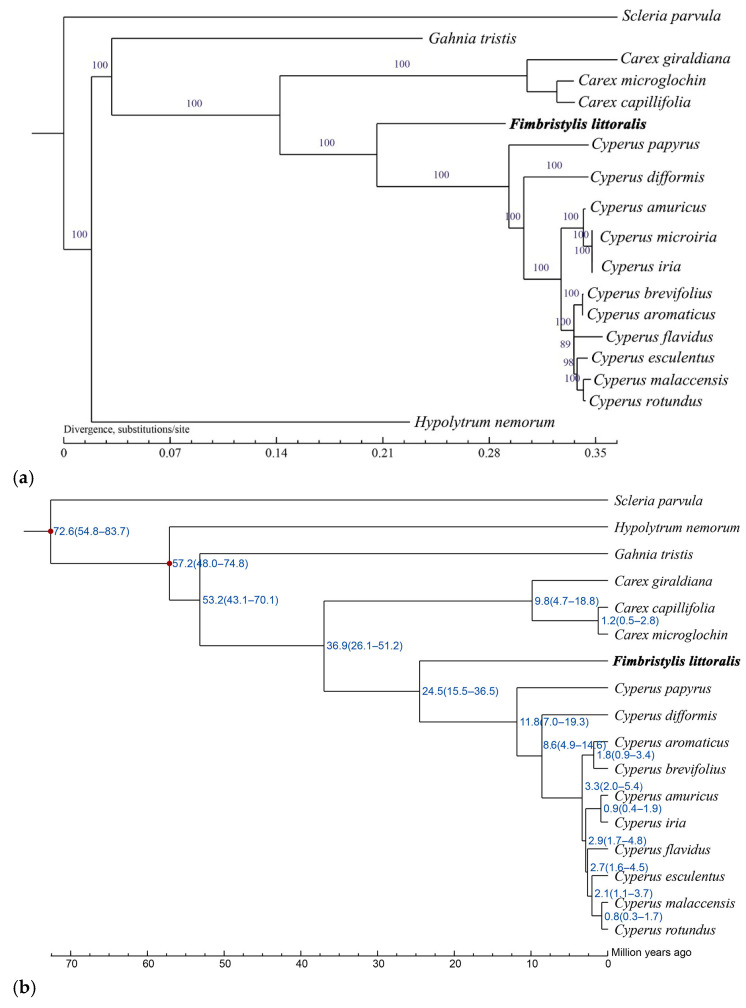
Phylogenetic and divergence analysis for 18 species of the family Cyperaceae. (**a**) Phylogenetic trees using maximum likelihood (ML), based on alignments of complete chloroplast genomes. (**b**) Divergence times and topologies of maximum likelihood trees based on the complete chloroplast genome.

**Figure 6 ijms-26-02321-f006:**
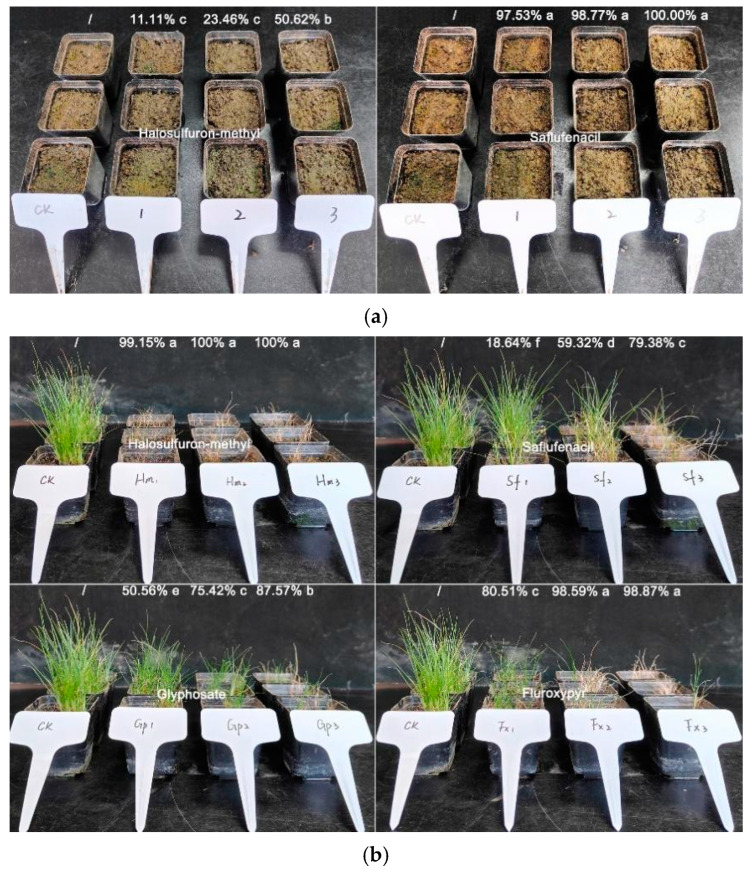
Sensitivity analyses for response of *Fimbristylis littoralis* to different herbicides. (**a**) Pre-emergence herbicides. a–c represents the significant difference of the inhibition rate of fresh weight among all herbicide treatments. (**b**) Post-emergence herbicides. a–f represents the significant difference of the inhibition rate of fresh weight among all herbicide treatments.

**Table 1 ijms-26-02321-t001:** List of genes encoded by chloroplast genome of *Fimbristylis littoralis*.

Category	Gene Groups	Gene Name
Photosynthesis	Subunits_of_photosystem_I	*psaA*, *psaB*, *psaC*, *psaI*, *psaJ*
	Subunits_of_photosystem_II	*pbf1*, *psbA*, *psbB*, *psbC*, *psbD*, *psbE*, *psbF*, *psbH*, *psbI*, *psbJ*, *psbK*, *psbL*, *psbM*, *psbT*, *psbZ*
	Subunits_of_NADH_dehydrogenase	*ndhA*, *ndhB*, *ndhC*, *ndhD*, *ndhE*, *ndhF*, *ndhG*, *ndhH*, *ndhI*, *ndhJ*, *ndhK*
	Subunits_of_cytochrome_b/f_complex	*petA*, *petB*, *petD*, *petG*, *petL*, *petN*
	Subunits_of_ATP_synthase	*atpA*, *atpB*, *atpE*, *atpF*, *atpH*, *atpI*
	Large_subunit_of_Rubisco	*rbcL*
Self-replication	Large_subunits_of_ribosome	*rpl14*, *rpl16*, *rpl2*, *rpl20*, *rpl22*, *rpl23*, *rpl32*, *rpl33*, *rpl36*
	Small_subunits_of_ribosome	*rps11*, *rps12*, *rps14*, *rps16*, *rps18*, *rps19*, *rps2*, *rps3*, *rps4*, *rps7*, *rps8*
	DNA-dependent_RNA_polymerase	*rpoA*, *rpoB*, *rpoC1*, *rpoC2*
	Ribosomal_RNAs	*rrn16*, *rrn23*, *rrn4.5*, *rrn5*
	Transfer_RNAs	*trnA-UGC*, *trnC-GCA*, *trnD-GUC*, *trnE-UUC*, *trnF-GAA*, *trnG-GCC*, *trnG-UCC*, *trnH-GUG*, *trnI-CAU*, *trnI-GAU*, *trnK-UUU*, *trnL-CAA*, *trnL-UAA*, *trnL-UAG*, *trnM-CAU*, *trnN-GUU*, *trnP-UGG*, *trnQ-UUG*, *trnR-ACG*, *trnR-UCU*, *trnS-GCU*, *trnS-GGA*, *trnS-UGA*, *trnT-UGU*, *trnV-GAC*, *trnV-UAC*, *trnW-CCA*, *trnY-GUA*, *trnfM-CAU*
Other genes	Maturase	*matK*
	Protease	*clpP1*
	Envelope_membrane_protein	*cemA*
	Acetyl-CoA_carboxylase	
	C-type_cytochrome_synthesis_gene	*ccsA*
	Translation_initiation_factor	
	protochlorophillide_reductase_subunit	
Unknown Genes	Proteins_of_unknown_function	*ycf3*, *ycf4*

## Data Availability

Raw reads of cp genomes of *Fimbristylis littoralis* were deposited in the NCBI GenBank database (accession number: PRJNA1206234).

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
