# Peer review of "The Complete Chloroplast Genome and the Phylogenetic Analysis of Fimbristylis littoralis (Cyperaceae) Collected in Cherry Blossom Nursery"

_ijms, 2025, doi:10.3390/ijms26052321_

Round 1
Reviewer 1 Report
Comments and Suggestions for Authors
This paper is one of many similar ones on the topic that have flooded 'molecular' journals during the last five to seven years. All these papers are similar to each other, like oranges or eggs from the same box. This paper (like many similar ones) is technically sound and, of course, acceptable after minor revision. The major issue I see within this and similar research is the principal irrelevance of the discussion and conclusions to the content of the study. However, such an issue cannot be corrected in principle because it is now the law of the genre. So, I left it to the Editor to handle this issue. From the other side, the content of this paper (the sequenced genome) is important and must be published.
Minor (but serious):
- The individuals of what taxon have been investigated? Fimbristylis miliacea is "nomen rejiciendum" (F. miliacea (L.) Vahl, Enum. Pl. [Vahl] 2: 287 (1805), nom. rej. https://www.ipni.org/?q=Fimbristylis%20miliacea).
- The assembly strategy of the plastome of 'F. miliacea' must be discussed better (lines 97-97, and below). Why did the authors perform the De Novo assembly if numerous related genomes are available? See, for example, Figures 4 and 5 for the summary.
- What is the difference between "Other genes" and "Unknown genes" in the author's Table 1 (starting after line 134)? Why are the functions of the matK gene and clpP1 gene unknown (no category is listed for these genes in the table)?
- What does it mean: "Phylogenetic and divergence analysis for 18 species of the family Cyperus." (line 197)? Why not use the standard family name?
- What is the meaning of Figure 5 if only Cyperus contains at least 940 species? Why did the authors perform this analysis? Only because other similar papers contain such analyses?
Author Response
This paper is one of many similar ones on the topic that have flooded 'molecular' journals during the last five to seven years. All these papers are similar to each other, like oranges or eggs from the same box. This paper (like many similar ones) is technically sound and, of course, acceptable after minor revision. The major issue I see within this and similar research is the principal irrelevance of the discussion and conclusions to the content of the study. However, such an issue cannot be corrected in principle because it is now the law of the genre. So, I left it to the Editor to handle this issue. From the other side, the content of this paper (the sequenced genome) is important and must be published.
Re: Thank you very much for your positive feedback on our research content and for the valuable suggestions aimed at improving the quality of our manuscript. We have carefully addressed each of your comments and made the necessary revisions accordingly. We hope that the revised manuscript now meets your expectations.
Minor (but serious):
- The individuals of what taxon have been investigated? Fimbristylis miliacea is "nomen rejiciendum" (F. miliacea (L.) Vahl, Enum. Pl. [Vahl] 2: 287 (1805), nom. rej. https://www.ipni.org/?q=Fimbristylis%20miliacea).
Re: Thank you very much for your professional comments! The correct Latin name should now be Fimbristylis littoralis. I have thoroughly reviewed and revised the entire manuscript, including both the textual content and the figures.
- The assembly strategy of the plastome of 'F. miliacea' must be discussed better (lines 97-97, and below). Why did the authors perform the De Novo assembly if numerous related genomes are available? See, for example, Figures 4 and 5 for the summary.
Re: Thank you for your comment. This is a good question. The de novo assembly strategy was adopted because, at the time of conducting the chloroplast genome sequencing, no genomic data for this species had been publicly released. Therefore, for this study, which represents the first chloroplast genome sequencing of this species, the de novo assembly strategy became the only viable approach. I have added this rationale to the Materials and Methods section to provide a basis for the choice of assembly strategy in line 363-364.
- What is the difference between "Other genes" and "Unknown genes" in the author's Table 1 (starting after line 134)? Why are the functions of the matK gene and clpP1 gene unknown (no category is listed for these genes in the table)?
Re: Thank you for your comment. Generally speaking, in chloroplast gene annotation, "Other genes" typically include those with more specialized functions or those that are less studied. Their roles involve the maintenance, regulation, or other non-core metabolic activities of chloroplasts, apart from photosynthesis and self-replication. On the other hand, unknown genes are usually of a predictive nature, and their functions have not yet been clearly identified or understood by the scientific community.
The matK gene and the clpP1 gene are associated with maturase and protease formation, respectively. However, they are not directly involved in the currently known core functions of chloroplasts, such as photosynthesis and self-replication. As a result, they are categorized under the group of "Other genes."
- What does it mean: "Phylogenetic and divergence analysis for 18 species of the family Cyperus." (line 197)? Why not use the standard family name?
Re: Thank you very much for your reminder. This was a typographical error. We have now used the standard family name. "Cyperus" has been revised to "Cyperaceae" in line 206-207.
- What is the meaning of Figure 5 if only Cyperus contains at least 940 species? Why did the authors perform this analysis? Only because other similar papers contain such analyses?
Re: Thank you for your comment. According to current taxonomic research, the genus Cyperus comprises several hundred species distributed worldwide. The primary reason for selecting the species included in our manuscript is their representativeness and common occurrence in our study region. Additionally, by integrating information already registered in the NCBI database, we ultimately made a comprehensive decision to include them. Beyond this, we also took into account the diversity within each genus.
Reviewer 2 Report
Comments and Suggestions for Authors
I reviewed the manuscript “The Complete Chloroplast Genome and the Phylogenetic Analysis of Fimbristylis miliacea (Cyperaceae) collected in Cherry Blossom Nursery”.
Comments:
- Plagiarism Percentage match: 46%, need to reduce
- L17, Chloroplast (Cp) and L18 and the whole manuscript (MS), the author used cp, use one format
- L23, the LSC / IRb junction. Check the space
- L25 and whole MS used g ai ha-1, mean?
- L85-89, revise and clearly explain the objectives
- L96, Figure S1? I have not found the supplementary figures
- L107-111, 149 simple sequence repeats (SSRs) were identified, including 23 distributed in coding region. According to the number of repeated bases, the type of SSR was shown in Figure 2a. There were 1932 long repeats (LR) in the cp genome of globe fringerush. According to the Hamming Distance (HD), the type of LR was shown in Figure 2b. I haven’t found the in figure 2a 149 simple sequence repeats (SSRs) and 1932 long repeats (LRs) in figure 2b. Statement and figures presentation is different.
- All figure and table legends need a dot (.), e.g., Figure 1.
- All figures quality is not good
- Figure 3: check X-axis
- In the discussion, material, and methods sections, the author cites Supplementary Tables, but there are no supplementary files. I have not found supplementary tables and figures.
- There are no legends of supplementary files
- L396 need space
- check the whole MS; some places need space, and some places do need space
- Lack of references in material and methods; need some references
The English could be improved to more clearly express the research
Author Response
I reviewed the manuscript “The Complete Chloroplast Genome and the Phylogenetic Analysis of Fimbristylis miliacea (Cyperaceae) collected in Cherry Blossom Nursery”.
Re: Thank you very much for your efforts in improving the quality of our manuscript. We have carefully revised the paper point by point according to your comments. We hope that the revised manuscript now meets your expectations.
- Plagiarism Percentage match: 46%, need to reduce
Re: Thank you for your comment. Due to the technical similarities in sections such as the research methodology, the manuscript is prone to being flagged for high repetition rates. We have carefully polished the manuscript to reduce the repetition rate.
- L17, Chloroplast (Cp) and L18 and the whole manuscript (MS), the author used cp, use one format
Re: Thank you very much for your reminder. We have reviewed the entire manuscript and made corrections to ensure a consistent format, using "cp".
- L23, the LSC / IRb junction. Check the space
Re: Thank you very much for your reminder. We have removed the unnecessary spaces to ensure they are presented in the correct format in line 23.
- L25 and whole MS used g ai ha-1, mean?
Re: Thank you very much for your reminder. The "g ai ha-1" is the unit for herbicide dosage, representing the amount of active ingredient applied per hectare. We have added an explanation where it first appears in the manuscript in line 26 and 215.
- L85-89, revise and clearly explain the objectives
Re: Thank you very much for your comments. We have rewritten this paragraph and hope it meets your requirements in line 96-103.
- L96, Figure S1? I have not found the supplementary figures
Re: Thank you for your comment. Figure S1 was indeed submitted, and I am not entirely sure why it was not displayed to you. I am now pasting it here for your review.
- L107-111, 149 simple sequence repeats (SSRs) were identified, including 23 distributed in coding region. According to the number of repeated bases, the type of SSR was shown in Figure 2a. There were 1932 long repeats (LR) in the cp genome of globe fringerush. According to the Hamming Distance (HD), the type of LR was shown in Figure 2b. I haven’t found the in figure 2a 149 simple sequence repeats (SSRs) and 1932 long repeats (LRs) in figure 2b. Statement and figures presentation is different.
Re: Thank you for your reminder. We have added the numbers of various types of SSRs and LRs to Figures 2a and 2b, respectively, to ensure consistency between the text and the figures.
- All figure and table legends need a dot (.), e.g., Figure 1.
Re: Thank you for your reminder. All the dots have been added.
- All figures quality is not good
Re: Thank you for your comment. We have improved the quality of all the figures. We hope our revisions meet your expectations.
- Figure 3: check X-axis
Re: Thank you for your reminder. We have reviewed and revised Figure 3 accordingly.
- In the discussion, material, and methods sections, the author cites Supplementary Tables, but there are no supplementary files. I have not found supplementary tables and figures.
Re: Thank you for your reminder. The supplementary files were indeed submitted, but I am unsure why they were not displayed to you. I have re-uploaded the supplementary files when submitting the revised manuscript. I am now providing screenshots of the supplementary files here for your review.
- There are no legends of supplementary files
Re: Thank you for your reminder. Legends have been added to all supplementary files.
- L396 need space
Re: Thank you very much for your reminder. The space has been added to the missing point at line 431.
- check the whole MS; some places need space, and some places do need space
Re: Thank you for your reminder. We have thoroughly checked the entire manuscript, added necessary spaces and removed unnecessary ones. All changes have been highlighted in the manuscript using track changes.
- Lack of references in material and methods; need some references
Re: Thank you for your reminder. We have added appropriate references to the Materials and Methods section.
Round 2
Reviewer 2 Report
Comments and Suggestions for Authors
The author addressed all my comments and suggestions. I haven't seen the supplementary files.